# A Global Evaluation of the Performance Indicators of Colorectal Cancer Screening with Fecal Immunochemical Tests and Colonoscopy: A Systematic Review and Meta-Analysis

**DOI:** 10.3390/cancers14041073

**Published:** 2022-02-21

**Authors:** Hanyue Ding, Jiaye Lin, Zijun Xu, Xiao Chen, Harry H. X. Wang, Liwen Huang, Junjie Huang, Zhijie Zheng, Martin C. S. Wong

**Affiliations:** 1JC School of Public Health and Primary Care, Faculty of Medicine, The Chinese University of Hong Kong, Hong Kong 999077, China; hanyue_ding@link.cuhk.edu.hk (H.D.); jiayelynn@link.cuhk.edu.hk (J.L.); zijunxu@link.cuhk.edu.hk (Z.X.); chenxiao9410@zju.edu.cn (X.C.); jason.huang@anfaclinic.cn (L.H.); junjie_huang@link.cuhk.edu.hk (J.H.); 2School of Public Health, Sun Yat-sen University, Guangzhou 510080, China; wanghx27@mail.sysu.edu.cn; 3Usher Institute, Deanery of Molecular, Genetic & Population Health Sciences, The University of Edinburgh, Edinburgh EH8 9AG, UK; 4School of Public Health, Peking University Health Science Center, Beijing 100191, China; zhengzj@bjmu.edu.cn; 5School of Public Health, Peking Union Medical College Hospital, Beijing 100730, China

**Keywords:** colorectal neoplasm, early detection of cancer, performance indicator

## Abstract

**Simple Summary:**

This study is a systematic review and meta-analysis of international literature on the achievements of various performance indicators for colorectal cancer screening programs. We systematically summarized performance indicators of organized colorectal cancer screening that used fecal immunochemical test as a primary screening modality, and colonoscopy as a subsequent confirmatory test, from 93 studies involving nearly 90 million people-times, and reported their pooled achievements based on random-effects models. We also performed meta-regression and subgroup analyses to explore the heterogeneity. Our findings could help to identify the areas that could be improved and finally optimize the organized colorectal cancer screening programs.

**Abstract:**

(1) Background: To summarize the achievements of the performance indicators of colorectal cancer (CRC) screening programs that used the fecal immunochemical test (FIT) as a primary screening modality and colonoscopy as a subsequent confirmatory test. (2) Methods: PubMed, Ovid MEDLINE, Embase, and Cochrane were searched from inception to September 2020. We included original articles published in English, and performed hand searching for relevant national reports. We generated pooled achievement estimates of the performance indicators by “metaprop” (R software 3.6.3). Meta-regression analyses and subgroup analyses were also conducted. (3) Results: We included 93 studies involving nearly 90 million people-times. The participation rate ranged from 6.80% to 95.98%, which was associated with study type, continents, FIT number, age, and round. The pooled FIT invalid rate and positivity rate were 1.08% and 7.28%, respectively. The pooled estimates of FIT detection were 2.26% for adenoma, 1.26% for advanced adenoma, and 0.28% for CRC. In addition, only seven studies reported that their colonoscopy compliance rate reached 90% among 69 studies. The colonoscopy completion rate (21/40 studies > 95%) and the complication rate (18/27 studies < 0.5%) were acceptable. (4) Conclusions: Our findings could help to identify the areas that could be improved and finally optimize the CRC screening programs.

## 1. Introduction

Worldwide, colorectal cancer (CRC) is the third most common cancer, and it was responsible for 10.2% of all new cancer cases and 9.4% of cancer-related deaths in 2020 [1]. According to recent studies, guaiac-based fecal occult blood test (gFOBT), fecal immunochemical test (FIT), flexible sigmoidoscopy (FS), and colonoscopy were associated with a reduction of 14–16% [2], 22% [3], 28% [4], and 68% [5] of all CRC-related deaths, respectively. Among these screening tools, FIT had a higher participation rate and positivity rate compared to gFOBT in the CRC screening programs [6]. Therefore, there is an increasing trend of adopting FIT as a primary screening test in the CRC screening program [7,8]. The performance indicators of screening programs need to be carefully monitored for quality control. In Canada, a comprehensive set of performance indicators was used, such as participation rate, fecal test inadequacy rate, FIT positivity rate, positive predictive value (PPV), and attendance to follow-up colonoscopy [9]. However, the survival benefit of screening may need a long time to be realized, as it takes on average 10 years to prevent one CRC-related death for every 1000 patients screened [10]. Hence, achieving pre-defined targets of performance indicators is essential in screening programs to enhance their success.

A large body of systematic reviews have summarized the performance indicators of different screening tools in CRC screening and diagnosis. They mainly described the comparison between gFOBT and FIT [6], factors associated with program adherence [11] and diagnostic accuracy of FIT [12]. However, most of these reviews did not distinguish opportunistic screening from organized screening [6,11,12,13]. Very few studies pooled the achievement of commonly used performance indicators in organized CRC screening. Unlike opportunistic screening, organized screening is more regularly monitored to ensure high standard and quality [8]. Although a recent study analyzed the data on the performance of organized CRC screening programs, it was only limited to the European data derived from the second European screening report [14]. To the best of our knowledge, there is no study published including all organized CRC screening programs in the world. Therefore, this was the first systematic review on the performance of all eligible population-based CRC screening programs that used FIT as a primary screening test, and colonoscopy as a subsequent confirmatory test, in various countries, and we also conducted meta-regression analyses and subgroup analyses to identify factors such as study type, continents, and age groups.

## 2. Materials and Methods

This study is a systematic review and meta-analysis that was conducted in compliance with the Meta-analysis of Observational Studies in Epidemiology (MOOSE) and was registered on PROSPERO International prospective register of systematic reviews (CRD42020142617).

### 2.1. Literature Search

We searched PubMed, Ovid MEDLINE, Embase, and Cochrane for articles published in English, from their inception to 8 September 2020. The keywords and MeSH terms used are shown in Appendix A. Hand searching was also performed to identify the relevant national CRC screening reports. The International Agency for Research on Cancer (IARC) handbooks of CRC screening (Volume 17) and reviews of current CRC screening programs were examined for potentially screening reports [8,15,16,17].

### 2.2. Inclusion and Exclusion Criteria

Population-based CRC screening programs that used FIT as a primary screening test and colonoscopy as a confirmatory test were eligible in this review. The exclusion criteria included the following: (1) randomized controlled trial and cost-effectiveness analysis; (2) studies on opportunistic screening, screening restricted in clinics, and screening targeting at special groups (people who were disabled or suffered from chronic diseases); (3) programs that included the use of other screening tools (i.e., gFOBT, FS, CT, and fecal DNA); (4) studies that examined outcomes other than performance indicators, such as CRC incidence or mortality; and (5) literature reviews, abstracts, or articles in other languages. If there were identical populations or overlapped study periods in different studies, we included the articles with the largest sample size.

### 2.3. Data Extraction and Quality Assessment

The literature search was conducted by two independent qualified reviewers (J.L. and Z.X.) based on screening titles and abstracts. Disagreements were resolved by a third reviewer (H.D.). The full texts were reviewed and appraised if the title and abstract were identified as eligible for inclusion. We extracted data including first author, year of publication, study location, project period, number of registered subjects, number of screening participants, age range, screening round, screening tools, the number of FIT used, the cut-off value of FIT, the number of subjects who submitted valid and invalid FIT, the number of subjects who were tested positive FIT, the number of subjects who attended colonoscopy appointment, quality of bowel preparation, number of individuals who completed colonoscopy procedure, complications due to colonoscopy, the number of subjects identified as having adenoma (including non-advanced adenoma and advanced adenoma), advanced adenoma, and CRC. Discrepancies in data extraction were resolved by group discussion.

We evaluated the quality of included articles by the Appraisal of cross-sectional studies (AXIS) [18]. There are 20 components to record a “yes”, “no” or “don’t know” response for assessing the risk of bias. Because we only included programs using FIT and colonoscopy as the screening tools, which have been demonstrated as having high diagnostic accuracy [12,19], we omitted domains eight and nine on the assessment of the validity and reliability of the measurement in the AXIS. Discrepancies on the risk of bias assessment were resolved by the third reviewer.

### 2.4. Statistical Analysis

We included the participation rate, the FIT invalid rate, the FIT positivity rate, the adenoma/advanced adenoma/CRC detection rate of FIT, PPV for adenoma/advanced adenoma/CRC, the colonoscopy compliance rate, the adequate quality rate of bowel preparation, the colonoscopy completion rate, and the colonoscopy complication rate the as the outcome variables. Each performance indicator was defined according to what was specified in the guidelines [20,21]. Meanwhile, a random-effects model was used to pool the rates with proportions and 95% confidence interval (95% CI) that was adopted in a previous study for the synthesis of performance indicators [14]. Heterogeneity was assessed by I^2^ and I^2^ of 25%, 50%, and 75%, which indicated low, medium, and high heterogeneity, respectively [22]. Egger’s tests were used to evaluate publication bias and any *p* value < 0.05 was considered statistically significant. Meta-regression random-effects models were performed to investigate the heterogeneity of the main pooled estimates. Subgroup analyses were also conducted in terms of the type (article and report), sample size (<50,000, 50,000–500,000, and >500,000), continents (Asia Pacific, Europe, North America, and South America), the number of FIT used (one and two), age groups (>50, the program was started from the age of around 50 years; >55, the program was started from the age of around 55 years), the FIT cut-off value (<100 ng hemoglobin/mL buffer; 100 ng/mL; >100 ng/mL), and screening round (initial round: first screening round; subsequent round: following screening round involved all follow-up tests and initial tests of new participants). All statistical analyses were performed by R software (version 3.6.3) and the function “metaprop” was adopted to conduct the meta-analysis of rates to generate pooled estimates.

## 3. Results

A total of 12,253 citations were identified from the literature search. A total of 7721 citations remained after the removal of duplicates, of which 7380 citations were further excluded in the first stage of screening based on titles or abstracts. In the second stage of screening, 341 full-text articles were assessed for eligibility and 22 reports were collected from national websites by hand searching. After the comprehensive review, a total of 93 studies from 63 articles and 22 reports were included in this meta-analysis (Figure 1).

### 3.1. Study Characteristics and Quality Assessment

The characteristics of all eligible studies are presented in Appendix A. All studies were published between 1996 and 2020. Most CRC screening programs (75/93) were started from the age of around 50 years, and in 39 studies, the eligible participants were aged between 50 and 74/75 years. Furthermore, 53 studies involved one FIT sample for screening, while 26 studies supplied two FIT samples for participants. Most of the studies adopted quantitative FIT samples, as only 16 studies did not mention the brand or the cut-off value of FIT. Nearly half of the studies (46/93) set 100 ng hemoglobin per ml buffer as the cut-off value for quantitative FIT, which corresponded to 20 μg hemoglobin per g feces. The quality of included articles was assessed by AXIS and is shown in Appendix A.

### 3.2. Participation Rate

A total of 69 studies reported participation rate, ranging from 6.80% to 95.98% among more than 73 million person-times invited (Figure 2). The random-effects model showed that the overall participation rate was 54.00% (95% CI: 49.28–58.69%), whereas the heterogeneity was high (I^2^ = 100%; Table 1). Publication bias was found in the pooled participation rate (*p* < 0.05). In the univariate subgroup analyses, articles (59.28%, 95% CI: 53.74–64.71%) reported higher participation rate than reports (43.47%, 95% CI: 36.39–50.48%) (*p* < 0.001). The studies with a larger sample size reported the lower participation rates (sample size <50,000: 67.53%, 50,000–500,000: 48.79%, and >500,000: 45.37%, *p* = 0.005; Appendix A). The participation rates were 55.25% (95%CI: 46.90–63.45%) in Asia Pacific, 52.72% (95% CI: 48.82–56.60%) in Europe, 45.57% (95% CI: 35.89–55.41%) in North America, and 90.19% (95% CI: 89.90–90.49%) in South America. A higher participation rate was found in screening with one FIT test (58.71%, 95% CI: 54.59–62.77%), initial round screening (63.97%, 95% CI: 59.53–68.30%), and older age group (63.48%, 95% CI: 56.40–70.28%) than screening with two FITs (44.32%, 95% CI: 40.93–47.73%), subsequent round (45.82%, 95% CI: 41.36–50.32%), and younger age group (50.28%, 95% CI: 45.84–54.73%) (*p* < 0.001). In multivariate meta-regression, the study type, continents, FIT number, age, and round were associated with the participation rate, which could explain 86.98% of heterogeneity between studies (Table 2).

### 3.3. FIT Invalid Rate and Positivity Rate

Twenty-eight studies included the proportion of invalid FIT, and the pooled FIT invalid rate was 1.08% (95% CI: 0.87% to 1.31%), which showed significant publication bias. Seventy-four studies presented a FIT positivity rate that ranged from 1.09% to 30.01%. The pooled FIT positivity rate was 7.28% (95% CI: 6.81–7.76%), while the I^2^ was 99.9% (Table 1). The continents, FIT number, and cutoff value had significant correlations with the invalid and positivity rate (Table 2). The positivity rate with two FITs was higher than that with one FIT in the Asia Pacific (two FITs: 8.17%, 7.50–8.87%; one FIT: 4.34%, 2.10–7.33%), Europe (two FITs: 8.70%, 6.67–10.96%; one FIT: 5.39%, 4.84–5.97%), and South America (two FITs: 15.19, 14.74–15.65%; one FIT: 11.07%, 10.47–11.67%), while Europe, North America, and South America had a higher positivity rate than the Asia Pacific (*p* < 0.01). Meanwhile, the positivity rate showed a decreasing trend with an increasing cutoff value, which was more pronounced in screening with one FIT (Figure 3).

### 3.4. Detection Rate and PPV

The pooled detection rates for adenoma, advanced adenomas and CRC were 2.26% (95%CI: 2.00% to 2.53%), 1.26% (95% CI: 1.10% to 1.44%), and 0.28% (95% CI: 0.25–0.31%), respectively (Table 1). The adenoma detection rate was associated with cutoff value and age; the advanced adenoma detection rate was correlated with continents and age, while continents, FIT number, cutoff value, and age had associations with the CRC detection rate (Table 2). The elder group and lower cutoff value showed a higher adenoma detection rate than the younger group and higher cutoff value, respectively. Europe and North America had a higher advanced adenoma detection rate than the Asia Pacific. For CRC detection, both in Europe and North America, screening with two FITs had higher detection rates than screening with one FIT. Furthermore, a higher cut-off value showed a decreasing detection rate of CRC in screening with one FIT (Figure 4).

The pooled PPV for adenoma, advanced adenoma, and CRC was 44.79% (95% CI: 41.8–47.79%), 27.13% (95% CI: 24.39–29.97%), and 5.48% (95% CI: 4.96–6.02%), respectively (Table 1). After performing the multivariate meta-regression, we observed there was a positive association between age and PPV for adenoma. Moreover, the PPV for advanced adenoma and CRC were influenced by the continents. In contrast with the positivity rate and detection rate, a higher cutoff value was correlated with a higher PPV for advanced adenoma (Table 2).

### 3.5. Indicators Related to Colonoscopy

A total of 69 studies were included to estimate the colonoscopy compliance rate, involving 1.3 million participants. The wide variability of the compliance rate was 31.42–96.01%, and only seven studies reported that the compliance rate was higher than 90%. Fourteen studies described the quality of bowel preparation, four of which adopted the Boston Bowel Preparation Scale, and two studies used the original/modified Ottawa Scale. Only one study (65.20%) did not reach 85%. Among 40 studies that involved a completion rate, half of studies achieved 95% and three-quarter studies reached 90%. A total of 27 studies reported the complication rate (0.00–1.23%), 18 of which were less than 0.5% (Table 1).

## 4. Discussion

To the best of our knowledge, this is the first large-scale literature review that summarized quantitative data on the performance of organized CRC screening programs worldwide. Quality indicators of FIT and colonoscopy were based on definitions from guidelines, which confirm the comparability of all indicators. Meanwhile, all FIT-related indicators were pooled and we performed a meta-regression using the random-effects model to interpret the heterogeneity. Furthermore, our subgroup analyses adjusted the covariates of indicators according to significant associations.

Our overall results were similar to current benchmarks or evidence, while our findings showed a wider variability as this is a worldwide systematic review. Most included CRC screening programs used quantitative FIT rather than qualitative FIT, as the former allows for the adjustment of cut-off values tailored to a country’s colonoscopy resources [23]. A higher cut-off value is associated with a lower positivity, adenoma, and CRC detection, but a higher PPV for advanced adenoma. A previous meta-analysis has proven that a lower cutoff value improves the sensitivity, but there is a corresponding decrease in specificity [12]. Therefore, more false-positive samples caused lower PPV for advanced adenoma with a lower cutoff value of FIT, which was consistent with other studies [20,24]. In order to balance the performance and health resources, half of the studies followed recommendations to set 20 μg/g as the FIT cut-off value. Meanwhile, although our results indicated that the FIT number indeed had an impact on the performance of CRC detection, the influence was very small (the adjusted coefficient was 0.013 of two FIT). Additionally, a Swedish study reported that screening with one FIT at 20 and 40 μg/g had better performance than screening with two FITs at 40 and 80 μg/g [25]. In consideration of the negative association between the FIT number and participation rate, we supported screening with one FIT rather than two FITs.

Continents and age groups also contributed to the heterogeneity between studies. According to the epidemiology of CRC, Europe and America had a higher incidence than the Asia Pacific [1], which led to the high FIT positivity and neoplasm detection. This was the same reason for the difference between age groups, as age is a risk factor of CRC. Due to the higher incidence in younger people, the United States Preventive Services Taskforce (USPSTF) recommends that CRC screening should start at 45 years [26]. Although we included three studies in which the age range was from 40/45 years, only one was performed in the last 20 years, hence the number was too small to show the synthesis indicator in the subgroup.

Most studies showed that the initial (first screening) tests had a better performance than the subsequent (screening performed by participants who have tested in the previous round) tests, [14,20] while in our study, the round had no significant association with most indicators, except the participation rate. This inconsistency could be induced by the definition of the screening round. The subsequent rounds extracted in our study involved all follow-up tests and initial tests of new participants. As we did not have the original data of the included studies, it was hard to break down data by initial and subsequent tests. Therefore, our covariates of the screening round had a lower impact on indicators than the screening history.

The colonoscopy compliance rate represents important determinants of the effectiveness of screening programs on CRC-related mortality reduction. The European guidelines recommended over 90% as the satisfactory rate of compliance with colonoscopy attendance in patients with positive fecal tests [27]. Furthermore, the United States Multi-Society Task Force (USMSTF) recommended that the colonoscopy compliance rate should reach 80% in the United States [28]. However, our compliance rate did not achieve the benchmarks. The quality of bowel preparations exerted a substantial impact on the efficiency of colonoscopy. Rex and colleagues recommended that the rate of adequate bowel preparation should be no less than 85% [21]. Our findings reported excellent results, but there were only six studies that adopted validated scales (Boston Bowel Preparation Scale/Ottawa Scale). The performance of cecal intubation is associated with interval CRC [29], hence the completion rate has been recommended as an important indicator, and 95% has been proposed as the target for screening colonoscopies [21,30]. Half of the studies reached the target, and thus evidence-based interventions should be taken to enhance the completion rate. These include colonoscopist training, optimization of bowel preparation and sedation, and the use of new endoscopic techniques [23]. Data from randomized controlled trials indicated severe complications were around 0.5–1.5% [31], and our results showed an acceptable value.

According to our results, the performance of global CRC screening showed large variability. In this case, a fixed benchmark may not provide enough evidence in consideration of the local conditions. Besides the overall estimates, we conducted several subgroup analyses to stratify the indicators, which may provide a specific reference for future CRC screening programs. The meta-regression in our study may help policymakers to consider the impact of factors and choose a suited screening strategy to balance the health resources and expected performance.

There are some limitations to the study. Firstly, we only included studies using FIT as the initial screening tool, followed by colonoscopy after a positive FIT, because in many studies, FIT was consistently proven to be better than gFOBT [6,32]. Furthermore, FIT has gradually become the most common test for initial CRC screening, and many countries have changed gFOBT to FIT in their programs [31,33]. Secondly, the economics of the countries and the development of health systems would also influence the performance of CRC screening, but we only adjusted some screening modalities to control the covariates in the meta-regression. Thirdly, there were some programs in Australia, Italy, and the Netherlands where the performance indicators were reported on a yearly basis. Finally, we only included articles presented in English.

## 5. Conclusions

Overall, we summarized the achievements of various indicators from population-based CRC screening programs using FIT and subsequent colonoscopy on a global scale, and explained the wide variability of these indicators in terms of screening modalities. These findings could help to identify the processes that can be improved and can finally optimize the CRC screening programs.

## Figures and Tables

**Figure 1 cancers-14-01073-f001:**
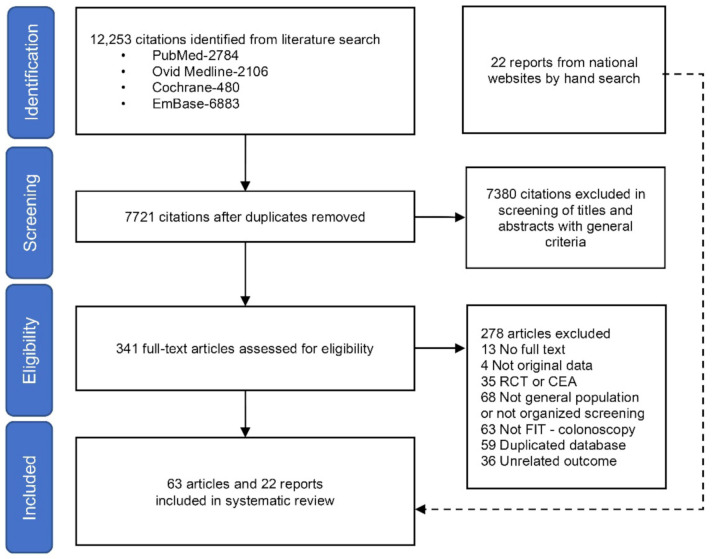
Flow diagram of this systematic review. RCT—randomized controlled trials; CEA—cost-effectiveness analyzes; FIT—fecal immunochemical test.

**Figure 2 cancers-14-01073-f002:**
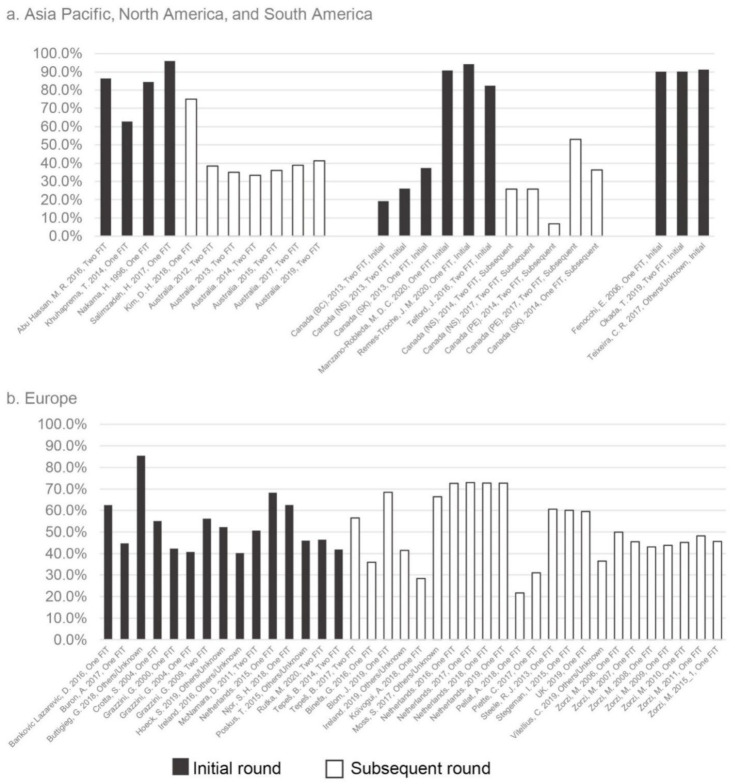
Participation rates by continents, screening round, and FIT number. (**a**) The participation rates in Asia, North America, and South America; (**b**) The participation rates in Europe.

**Figure 3 cancers-14-01073-f003:**
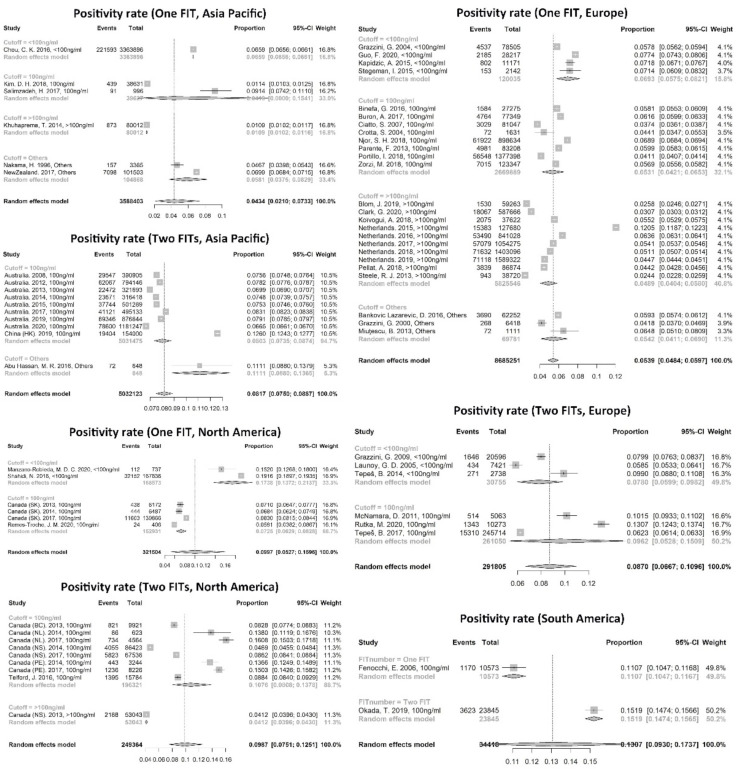
Positivity rates by continents, FIT number, and cutoff value.

**Figure 4 cancers-14-01073-f004:**
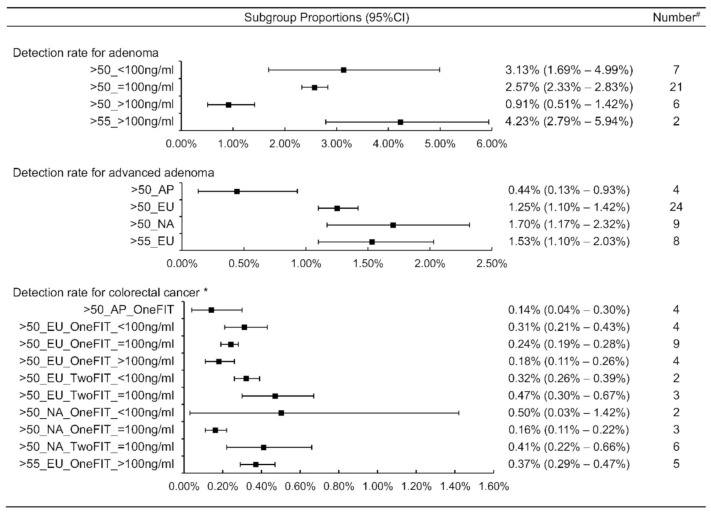
Stratified analyses of detection rates for adenoma, advanced adenoma and colorectal cancer. ^#^ We stratified subgroups to adjust detection rate until the number of studies was less than two. * The cut-off of 100 ng hemoglobin per ml of buffer solution was corresponding to 20 μg hemoglobin/g feces. AP—Asia pacific; EU—Europe; NA—North America.

**Table 1 cancers-14-01073-t001:** Estimates of performance indicators.

Performance Indicators	Studies	Number	Range	Estimates (95%CI)	I^2^
Participation rate	69	73,047,226	(6.80%, 95.98%)	54.00% (49.28–58.69%)	100.0%
Invalid rate	28	10,695,371	(0.09%, 5.30%)	1.08% (0.87–1.31%)	99.9%
Positive rate	74	24,374,662	(1.09%, 30.01%)	7.28% (6.81–7.76%)	99.9%
Adenoma detection rate	49	17,613,793	(0.23%, 7.73%)	2.26% (2.00–2.53%)	99.9%
Advanced adenoma detection rate	47	20,943,205	(0.09%, 3.69%)	1.26% (1.10–1.44%)	99.9%
CRC detection rate	61	23,584,358	(0.00%, 1.16%)	0.28% (0.25–0.31%)	99.5%
PPV for adenoma	52	823,533	(11.11%, 72.68%)	44.79% (41.8–47.79%)	99.9%
PPV for advanced adenoma	49	920,387	(6.67%, 48.36%)	27.13% (24.39–29.97%)	99.9%
PPV for CRC	61	973,245	(0.0%, 15.07%)	5.48% (4.96–6.02%)	99.1%
Colonoscopy compliance rate	69	1,310,390	(31.42%, 96.01%)	7/69 studies >90%, 30/69 studies > 80%	
Bowel preparation (adequate)	14	202,936	(65.20%, 97.30%)	13/14 studies > 85%	
Colonoscopy completion rate	40	798,029	(73.16%, 100.00%)	21/40 studies > 95%, 30/40 studies > 90%	
Colonoscopy complication rate	27	811,334	(0.00%, 1.23%)	18/27 studies < 0.5%	

CRC—colorectal cancer; PPV—positive predictive value.

**Table 2 cancers-14-01073-t002:** Meta-regression and subgroup analyses.

Performance Indicators	Meta-Regression	Significant Covariates ^a^						Adjusted R^2^
		Type	Continents	FIT Number	FIT Cutoff Value ^b^	Age Group	Screening Round	
Participation rate	Uni	√	√	√	√	√	√	-
	Multi	Article: ref **Report:** **−0.168 *****	AP: ref **Euro: −0.221 ***** **NA: −0.104 **** **SA: 0.266 *****	One: ref **Two:** **−0.135 *****	-	>50 yrs: ref **>55 yrs: 0.322 *****	Initial: ref **Subsequent:** **−0.099 *****	86.98%
Invalid rate	Uni	-	√	√	√	-	-	-
	Multi	-	Euro: ref **NA: 0.048 ****	One: ref **Two: 0.032 *****	<100 ng/mL: ref **>100 ng/mL: 0.048 ****	-	-	54.63%
Positive rate	Uni	√	√	√	√	√	√	-
	Multi	-	AP: ref **Euro: 0.04 **** **NA: 0.082 ***** **SA: 0.139 *****	One: ref **Two: 0.048 *****	<100 ng/mL: ref **100 ng/mL: −0.037 **** **>100 ng/mL: −0.083 *****	-	-	21.25%
FIT adenoma detection rate	Uni	√	√	√	√	√	-	-
	Multi	-	-	-	<100 ng/mL: ref **>100 ng/mL:** **−0.062 *****	>50 yrs: ref **>55 yrs: 0.063 *****	-	41.13%
FIT advanced adenoma detection rate	Uni	-	√	√	√	√	√	-
	Multi	-	AP: ref **Euro: 0.053 ***** **NA: 0.062 ***** **SA: 0.044 ***	-	-	>50 yrs: ref **>55 yrs: 0.029 *****	-	50.46%
FIT CRC detection rate	Uni	-	√	√	√	√	√	-
	Multi	-	AP: ref **Euro: 0.013 ***** **NA: 0.012 *** **SA: 0.054 *****	One: ref **Two: 0.013 *****	<100 ng/mL: ref **>100 ng/mL: −0.009 ***	>50 yrs: ref **>55 yrs: 0.015 *****	-	56.11%
PPV for adenoma	Uni	√	√	-	-	√	√	-
	Multi	-	-	-	-	>50 yrs: ref **>55 yrs: 0.112 ***	-	25.56%
PPV for advanced adenoma	Uni	-	√	√	√	√	-	-
	Multi	-	AP: ref **Euro: 0.130 ***** **NA: 0.090 ***	-	<100 ng/mL: ref **100 ng/mL: 0.087 **** **>100 ng/mL: 0.088 ****	-	-	63.85%
PPV for CRC	Uni	-	√	-	√	√	-	-
	Multi	-	AP: ref **SA: 0.011 ***	-	-	-	-	54.83%

Boldface indicates statistical significance (* *p* < 0.05, ** *p* < 0.01, *** *p* < 0.001). ^a^ Significant covariates in univariate meta-regression (√) were included in multivariate meta-regression analyses. ^b^ The cut-off of 100 ng hemoglobin per mL of buffer solution was corresponding to 20 μg hemoglobin/g feces. FIT—fecal immunochemical test; CRC—colorectal cancer; PPV—positive predictive value; AP—Asia Pacific; Euro—Europe; NA—North America; SA—South America.

## Data Availability

The datasets used and analyzed during the current study are available in the figshare (https://doi.org/10.6084/m9.figshare.15059754, accessed on 17 January 2022).

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
