# Peer review of "A Global Evaluation of the Performance Indicators of Colorectal Cancer Screening with Fecal Immunochemical Tests and Colonoscopy: A Systematic Review and Meta-Analysis"

_cancers, 2022, doi:10.3390/cancers14041073_

Round 1

Reviewer 1 Report

This article presents a systematic review and meta-analysis of the literature on performance indicators of CRC screening programmes that used fecal immunohistochemical test (FIT) as a primary screening modality and colonoscopy as a subsequent  confirmatory test. The strength of the study is that in contrast to previous similar articles, this one included CRC screening programmes in various countries of the world and not just European ones. As such, it provides interesting information and could be of help to improve and optimize the organized CRC screening programmes.

I have only one minor suggestion on how to improve the quality of the paper: the first paragraph (lines 231-234) of the discussion is redundant, it is not directly related to the content of this article. It seems rather as a general advice on how to write a paper. I suggest to delete it.

Reviewer 2 Report

In this manuscript, Ding et al., review and analyzed international literature on performance indicators for colorectal cancer (CRC) screening programs, specifically programs that used FIT as a primary screening test and colonoscopy as a subsequent confirmatory test. They performed meta-regression and subgroup analysis to identify the large variability and heterogeneity, since the analysis has been conducted for CRC screening programs worldwide.

In general, the manuscript is well written and the research is ambitious. The Methods section is clear. Lines 231-234 should be removed as they refer tow writing instructions for the manuscript . The authors are aware of the limitations of the study and I would only suggest to include further discussion on how might the findings impact in clinical practice for CRC screening programs. 

Reviewer 3 Report

In this manuscript Ding and colleagues make a systematic review and meta-analysis of performance characterisits of FIT-based CRC screening. The manuscript is relevant and interesting, updating the state of the art of this topic. Some concerns should be raised:

1: Title. Must make reference to FIT-based CRC screening, as this is the focus of the study.

2. At the end of the introduction section the authors said: "...we also conductesd meta-regression analyses and subgroup analyses to identify the factors". It is not clear which factors are the authors talking about.

3. Results, participation rate: I'm not sure if participation rate is expressing real data. Some of the participation rates are including reports of small pilot studies, with very high participation rates. I would suggest to separate these reports from small pilot studies and large series from established screening programs. These last are really exploring which is the participation rate in screening programs, having highest interest.

4. Figure 4. I don't fully understand this figure, what is the meaning of ther "Y" axis. Please, make it more clear. 

5. First paragraph of the discussion is inadequate. It looks like a guide for outlining the discussion. 
